# miR-4432 Targets FGFBP1 in Human Endothelial Cells

**DOI:** 10.3390/biology12030459

**Published:** 2023-03-16

**Authors:** Roberta Avvisato, Pasquale Mone, Stanislovas S. Jankauskas, Fahimeh Varzideh, Urna Kansakar, Jessica Gambardella, Antonio De Luca, Alessandro Matarese, Gaetano Santulli

**Affiliations:** 1Division of Cardiology, Department of Medicine, Albert Einstein College of Medicine, New York, NY 10461, USA; 2Wilf Family Cardiovascular Research Institute, Albert Einstein College of Medicine, New York, NY 10461, USA; 3Department of Advanced Biomedical Sciences, “*Federico II*” University, 80131 Naples, Italy; 4Department of Mental and Physical Health and Preventive Medicine, University of Campania “*Luigi Vanvitelli*”, 81100 Caserta, Italy; 5“*Antonio Cardarelli*” Hospital, 80100 Naples, Italy; 6Department of Molecular Pharmacology, Albert Einstein College of Medicine, New York, NY 10461, USA; 7Fleischer Institute for Diabetes and Metabolism (*FIDAM*), New York, NY 10461, USA; 8Einstein-Mount Sinai Diabetes Research Center (*ES-DRC*), Albert Einstein College of Medicine, New York, NY 10461, USA

**Keywords:** blood–brain barrier, blood pressure, cerebrovascular disease, endothelial dysfunction, hBMECs, hypertension, HUVEC, microRNA, miRNA, miR-4432-3p

## Abstract

**Simple Summary:**

The inner layer of blood vessels is formed by endothelial cells. When these cells do not work properly, several issues ensue in the human body. One of these issues is elevated blood pressure, also known as hypertension, which is an established risk factor for ischemic heart disease, stroke, chronic kidney disease, and dementia. However, the exact mechanisms linking dysfunctional endothelium and hypertension are not fully defined. In this work, we discovered that a small nucleic acid (miR-4432) is able to target and inhibit a specific gene (fibroblast growth factor binding protein 1, *FGFBP1*) in human brain microvascular endothelial cells, and we demonstrate for the first time that this miR-4432 significantly reduces endothelial oxidative stress, a well-established feature of hypertension. Taken together, our findings provide unprecedented mechanistic insights and open the field to new studies aimed at ameliorating endothelial dysfunction by harnessing miR-4432-based strategies.

**Abstract:**

MicroRNAs (miRs) are small non-coding RNAs that modulate the expression of several target genes. Fibroblast growth factor binding protein 1 (FGFBP1) has been associated with endothelial dysfunction at the level of the blood–brain barrier (BBB). However, the underlying mechanisms are mostly unknown and there are no studies investigating the relationship between miRs and FGFBP1. Thus, the overarching aim of the present study was to identify and validate which miR can specifically target FGFBP1 in human brain microvascular endothelial cells, which represent the best in vitro model of the BBB. We were able to identify and validate miR-4432 as a fundamental modulator of FGFBP1 and we demonstrated that miR-4432 significantly reduces mitochondrial oxidative stress, a well-established pathophysiological hallmark of hypertension.

## 1. Introduction

Hypertension is a leading risk factor for ischemic heart disease, stroke, chronic kidney disease, and dementia [1]. It is a multifactorial disease involving interactions among genetic, environmental, demographic, vascular, and neuroendocrine factors [2,3]. Endothelial dysfunction is an established hallmark of hypertension [4,5,6]; however, the exact molecular mechanisms linking dysfunctional endothelial cells (ECs) and high blood pressure are not fully understood. Several genome-wide association studies (GWAS) have identified a number of genes associated with hypertension [7,8], but only a few of these genes have been functionally validated. In 2019, the International Consortium of Antihypertensive Pharmacogenomics Studies (ICAPS) recognized fibroblast growth factor binding protein 1 (FGFBP1) as one of the genes involved in the regulation of blood pressure [9]. FGFBP1 is a key promoter of the development of the blood–brain barrier (BBB) [10], an aspect that is especially relevant considering that ECs are a major component of the BBB [11], which is crucial for maintaining neuronal and glial function [12]. Specifically, FGFBP1 has been implied in refining and maintaining barrier characteristics in the mature BBB endothelium [13].

MicroRNAs (miRs) are a relatively well conserved group of small (~21 nucleotides) non-coding RNAs that modulate the expression of their target genes: miRNAs can bind the 3′ untranslated region (UTR) of specific genes, thereby inhibiting their expression. Thus, miRNAs have been involved in numerous pathological and physiological processes [14,15]. Others and ourselves have, in the last decades, identified a variety of miRs involved in the regulation of endothelial function [16,17].

Since FGFBP1 has been previously linked to the modulation of the BBB, and precisely to endothelial function, the central scope of the present study is to detect which miR can target FGFBP1 in hBMECs (human brain microvascular endothelial cells).

## 2. Results

### 2.1. miR-4432 Targets FGFBP1 in a Conservative Manner

We applied bioinformatic analyses and functional experiments which led, for the first time to our knowledge, to the identification of hsa-miR-4432-3p (miR-4432) as a crucial modulator of FGFBP1 transcription, in a manner that is highly conserved across different species, including primates such as chimpanzee (*Pan troglodytes*), orangutan (*Pongo abelii*), macaque (*Macaca mulatta*), and gorilla (*Gorilla gorilla*), although it is not detected in mouse (*Mus musculus*) and rat (*Rattus norvegicus*), as shown in Figure 1.

Furthermore, we designed a mutant construct of FGFBP1 3′-UTR (“FGFBP1 MUT”) that harbors nucleotide substitutions at the level of the miR-4432 binding sites of FGFBP1 3′-UTR, as illustrated in Figure 2.

### 2.2. miR-4432 Regulates FGFBP1 Transcription in Endothelial Cells

We first verified that miR-4432 is actually expressed in two different types of endothelial cells, namely hBMECs, which remain the best in vitro model of the BBB [18], and human umbilical vascular endothelial cells (HUVECs), and that its expression is regulated by miR-4432 mimic and miR-4432 inhibitor, as shown in Figure 3.

Then, we performed a series of experiments in hBMECs to test whether miR-4432 is a regulator of FGFBP1 transcription. Through luciferase assays, we determined that FGFBP1 is a target of miR-4432 (Figure 4); these findings were also endorsed in HUVECs (Appendix A).

### 2.3. FGFBP1 Expression Is Controlled by miR-4432

As depicted in Figure 5, we experimentally proved that miR-4432 significantly diminishes the mRNA expression of FGFBP1 in hBMECs.

These findings were then confirmed by immunoblot at the protein level (Figure 6), as well.

### 2.4. miR-4432 Regulates Mitochondrial Oxidative Stress in Human ECs

The next logical step was to gain more insights into the physiological and disease-related consequences of the interaction between miR-4432 and FGFBP1. The generation of mitochondrial reactive oxygen species (ROS) induced by the known vasoconstrictor angiotensin II (Ang II) in ECs [19] has been mechanistically implied in the pathogenesis of hypertension [20,21,22] and previous investigations have evidenced that the upregulation of FGFBP1 can increase oxidative stress signaling, leading to pro-hypertensive effects [23].

On these grounds, we quantified, by MitoSOX, the ROS production induced by Ang II in hBMECs transfected with miR-4432 mimic, miR-4432 inhibitor, or, as control, miR-scramble. Strikingly, we observed that mitochondrial oxidative stress was significantly reduced by miR-4432 mimic and increased by miR-4432 inhibitor (Figure 7).

To mechanistically prove the functional role of FGFBP1, we repeated the ROS quantification after the knock-down of FGFBP1, showing that in the absence of FGFBP1 there is no significant effect of miR-4432 (Figure 8).

## 3. Discussion

The experimental observation herein reported indicates that miR-4432 targets FGFBP1 in human ECs, representing a novel potential strategy against numerous diseases characterized by endothelial dysfunction, including hypertension [24,25,26,27,28,29].

Consistent with our results, hypertensive patients have been shown to have approximately 1.5- and 1.4-fold higher expression of FGFBP1 mRNA and protein compared to normotensive subjects [30], further corroborating the crucial role of FGFBP1 in the pathophysiology of hypertension.

A genetic polymorphism in the human FGFBP1 gene has been associated with a higher gene expression and an increased risk of familial hypertension [30]. Preclinical studies in spontaneously hypertensive rats substantiated a contribution of the FGFBP1 genomic locus to hypertension and to glomerular damage [31]. In addition, the induction of FGFBP1 in a transgenic mouse model resulted in sustained hypertension and increased vascular sensitivity to the vasoconstrictor angiotensin II (Ang II) via ROS and MAP kinase pathway signaling [23,32]. Taken together, these pieces of evidence indicate that FGFBP1 can finely control steady-state blood pressure, most likely by regulating vascular sensitivity to endogenous Ang II.

Another study explored the indirect relationship between FGFBP1 and miRs in human umbilical vein ECs, showing that miR-146a promotes angiogenesis by increasing FGFBP1 expression via targeting CREB3L1 (Cyclic AMP Responsive-Element-Binding Protein-3-Like 1) [33]. In agreement with these data, FGFBP1 has been shown to be significantly upregulated in the hemolytic uremic syndrome associated with human immunodeficiency virus (HIV-HUS), which is characterized by endothelial damage and microcystic tubular dilation [34,35]; furthermore, the inhibition of FGFBP1 was shown to be beneficial in preventing brain vessel damage triggered by acute kidney injury [32].

Intriguingly, FGFBP1 is also expressed in keratinocytes, infiltrating mononuclear cells, and Kaposi’s Sarcoma spindle cells [36,37]; its activation during the process of wound healing in the skin can induce angiogenic lesions that closely resemble Kaposi’s Sarcoma [36]. Equally importantly, FGFBP1 can promote hepatocellular carcinoma metastasis [38], and patients with pancreatic cancer who express higher FGFBP1 levels have been shown to have a worse prognosis [39]. 

So, FGFBP1 is generally considered an indicator of early stages of pancreatic and colorectal adenocarcinoma [40], and as a biomarker it is very useful in predicting bacillus Calmette–Guérin response in bladder cancer [41]. It has been shown to be significantly upregulated in early dysplastic lesions of the human colon as well as in primary and metastatic colorectal cancers, whereas its knock-down led to anti-proliferative effects [42,43,44]. Therefore, its targeting using miR-based approaches could also lead to novel strategies in oncology.

Last but not least, the FGF signaling pathway has been shown to be intimately involved in the regulation of the vascular tone, with important roles in a number of homeostatic processes including blood pressure regulation, inflammation, shock, and ischemia-reperfusion, as well as injury/repair situations involving the vasculature, nervous system and dermal wound healing [45,46], and it also affects vascular morphogenesis of pre-endothelial cells of the embryo [47]. One of the main limitations of our study is having performed just in vitro assays; however, the FGFBP1 targeting by miR-4432 was confirmed in two different cell types (i.e., hBMECs and HUVECs). Additional studies are necessary to confirm the effects of miR-4432 in the pathobiology of hypertension and other cardiovascular and cerebrovascular disorders.

In summary, we established that FGFBP1 is expressed in ECs and that miR-4432 finely controls its expression levels both at the mRNA and protein level.

## 4. Methods

### 4.1. Cells and Other Reagents

hBMECs were purchased from Neuromics (Catalog code number: HEC02; Minneapolis, MN, USA). HUVECs were purchased from ThermoFisher Scientific (Catalog code number: C0035C; Waltham, MA, USA). Cells were cultured at early passages (3–7) under standard conditions (37 °C, 5% CO_2_), as previously described [48]. In some assays, the cells were transfected with *pcDNA3.1-FGFBP1* plasmids obtained from GenScript (Piscataway, NJ, USA). All other reagents were obtained from Merck (Darmstadt, Germany).

### 4.2. Identification of miR-4432 as a Modulator of FGFBP1

To ascertain which miRs could specifically target the 3′-UTR of FGFBP1, we harnessed Target Scan Human 8.0, as reported previously [48]. The effects of miR-4432 on FGFBP1 gene transcription were assessed in hBMECs cells through a luciferase-reporter that contained the 3′-UTR of the predicted miR interaction site, in both the WT and mutated forms. The mutant of FGFBP1 3′-UTR (FGFBP1-MUT, see Figure 1 and Figure 2), which contained substituted nucleotides in the region of the predicted miR-4432 binding-site of FGFBP1 3′-UTR, was designed via the NEBase Changer and Q5-site-directed mutagenesis kit (New England-Biolabs, Ipswich, MA, USA) as previously reported [48].

Using Lipofectamine-RNAiMAX (Thermo Fisher Scientific), hBMECs were transfected (66% transfection efficiency) with 0.05 μg of the 3′-UTR plasmid as well as miR-4432 mimic (a chemically synthesized double-stranded RNA that mimics endogenous miR-4432, MedChemExpress, Monmouth Junction, NJ, USA) or miR-4432 inhibitor (a steric blocking oligonucleotide that hybridizes with mature miR-4432 and inhibits its function, IDT, Coralville, IA, USA), or a negative control (non-targeting scramble, IDT), reaching a final concentration of 50 nMol/L [48]. Utilizing the Luciferase-Reporter Assay System (Promega, Madison, WI, USA), we quantified Firefly-and-Renilla luciferase activities forty-eight hours after the transfection, as previously described [48]. In some experiments, endothelial cells were transfected with shRNA-FGFBP1 or shRNA-scramble (Origene, Rockville, MD, USA), following the manufacturer’s instructions. TaqMan microRNA Assays (Thermo Fisher Scientific) were used to quantify mature miR-4432 using U18 as endogenous control, as described in the literature [16]. FGFBP1 expression was assessed via RT-qPCR as previously reported [48], normalizing to glyceraldehyde 3-phosphate dehydrogenase (GAPDH). The sequences of oligonucleotide primers (Merck, Darmstadt, Germany) are shown in Table 1.

### 4.3. Immunoblotting

Immunoblotting assays were performed as previously described and validated by our group [16,49]; the intensity of the bands was quantified using FIJI (“Fiji Is Just Image J”) software. The antibody for FGFBP1 was purchased from ThermoFisher Scientific (Catalog code number: PA5-77220); the antibody for β-Actin was purchased from abcam (Cambridge, MA, USA; Catalog code number: ab8229).

### 4.4. Mitochondrial ROS

Mitochondrial ROS generation was assessed using MitoSOX Red (catalog code number: #M36008; Thermo Fisher Scientific) in hBMECs cells treated with Ang II (400 nMol for 4 h), as previously described [50].

### 4.5. Statistical Analysis

All data were expressed as means ± standard error of the means (SEMs). The statistical analyses were carried out using GraphPad 9 (Dotmatics, San Diego, CA, USA). Statistical significance, set at *p* < 0.05, was tested using the non-parametric Mann–Whitney U test or a two-way ANOVA followed by Bonferroni multiple comparison test, as appropriate.

## 5. Conclusions

Taken together, our results indicate for the first time, to the best of our knowledge, that miR-4432 specifically targets the 3′UTR of FGFBP1, thereby representing a novel potential strategy against hypertension, cerebrovascular disease, and other disorders characterized by endothelial dysfunction.

## Figures and Tables

**Figure 1 biology-12-00459-f001:**
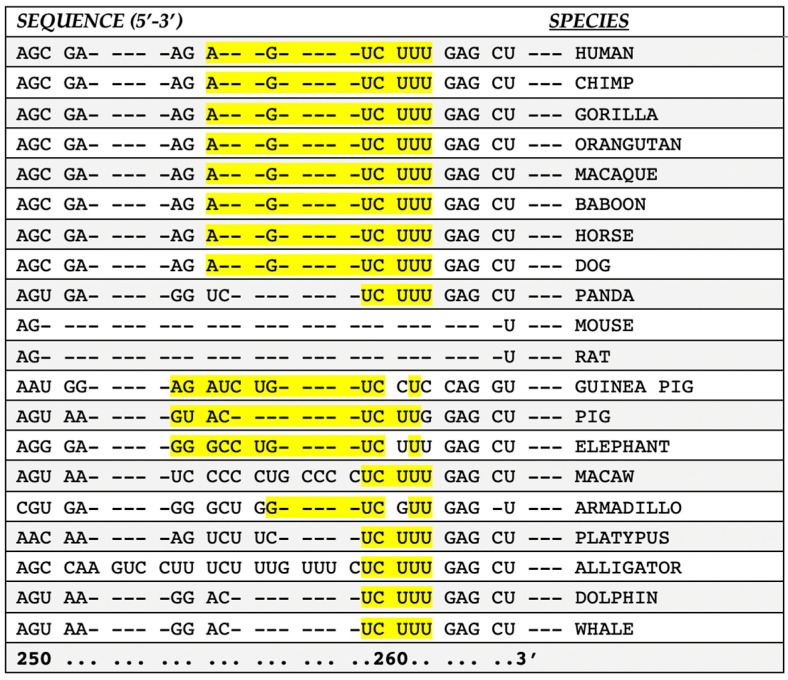
Identification of miR-4432 as a specific modulator of FGFBP1; the complementary nucleotides between the target region of FGFBP1 3′-UTR and hsa-miR-4432-3p, highlighted in yellow, are conserved across a number of different species, including primates.

**Figure 2 biology-12-00459-f002:**
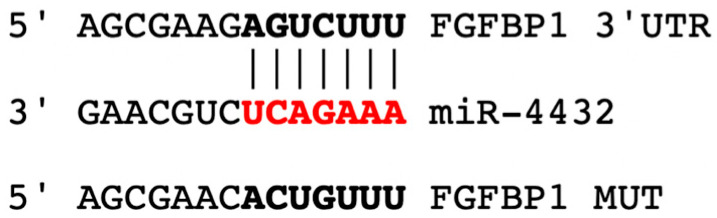
The designed mutant construct of FGFBP1 3′-UTR (FGFBP1 MUT) that harbors nucleotide substitutions at the level of the miR-4432 binding sites (indicated in red) of FGFBP1 3′-UTR, proving that miR-4432 specifically targets the 3′UTR of FGFBP1.

**Figure 3 biology-12-00459-f003:**
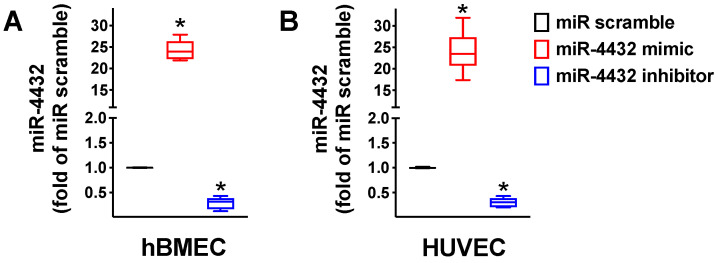
RT-qPCR showing that miR-4432 is expressed in both hBMECs (**A**) and HUVECs (**B**). All the assays were carried out in quadruplicate; the graphs indicate the median and the 5th to 95th percentiles; *: *p* < 0.01 vs. miR-scramble.

**Figure 4 biology-12-00459-f004:**
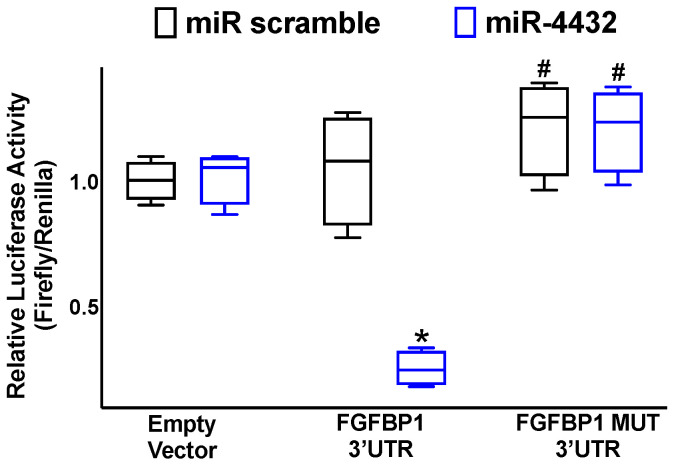
miR-4432 targets FGFBP1. Luciferase activity was quantified in hBMECs forty-eight hours after the transfection, utilizing the vector without FGFBP1 3′-UTR (“Empty Vector”), the vector that included the WT FGFBP1 3′-UTR (“FGFBP1 3′-UTR”), and the vector that included the mutated form of the FGFBP1 3′-UTR (“FGFBP1 MUT 3′UTR”); a miR-scramble (non-targeting miR) was used as an additional control. All the assays were carried out in quadruplicate; the graphs indicate the median and the 5th to 95th percentiles; *: *p* < 0.01 vs. miR-scramble; #: *p* < 0.05 vs. FGFBP1 3′UTR.

**Figure 5 biology-12-00459-f005:**
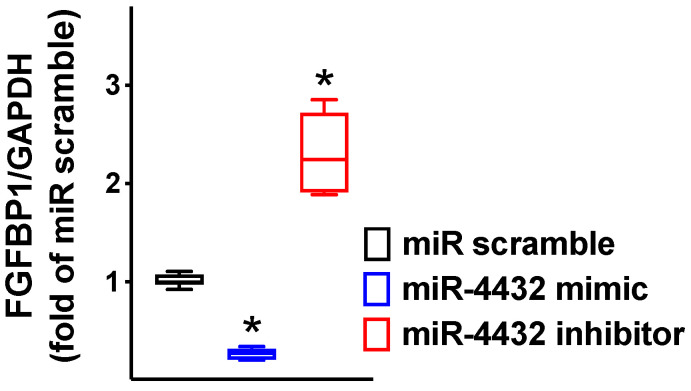
In hBMECs, FGFBP1 transcription was diminished by miR-4432 and augmented by miR-4432 inhibitor. FGFBP1 mRNA was quantified via RT-qPCR in hBMECs that had been transfected for forty-eight hours with the miRs indicated in the figure; values were normalized to GAPDH (glyceraldehyde-3-phosphate-dehydrogenase). All the assays were carried out at least in triplicate; the graph shows the medians and the 5th to 95th percentiles; *: *p* < 0.01 vs. miR-scramble. Sequences of the primers that have been used for the RT-qPCR are shown in Table 1.

**Figure 6 biology-12-00459-f006:**
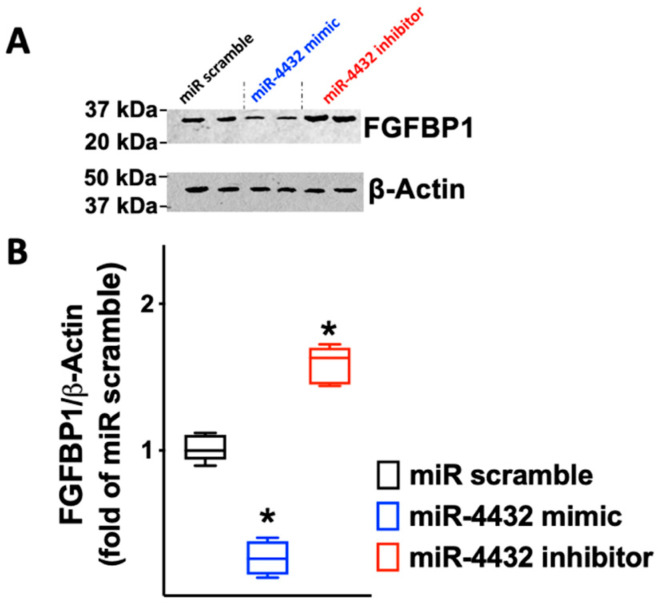
The observations detected by RT-qPCR in terms of mRNA were upheld by Western blots, as shown in the representative blots, showing two biological replicates per condition (**A**) and their quantification (**B**). All assays were carried out at least in triplicate; the graph represents the medians and the 5th to 95th percentiles; *: *p* < 0.01 vs. miR-scramble.

**Figure 7 biology-12-00459-f007:**
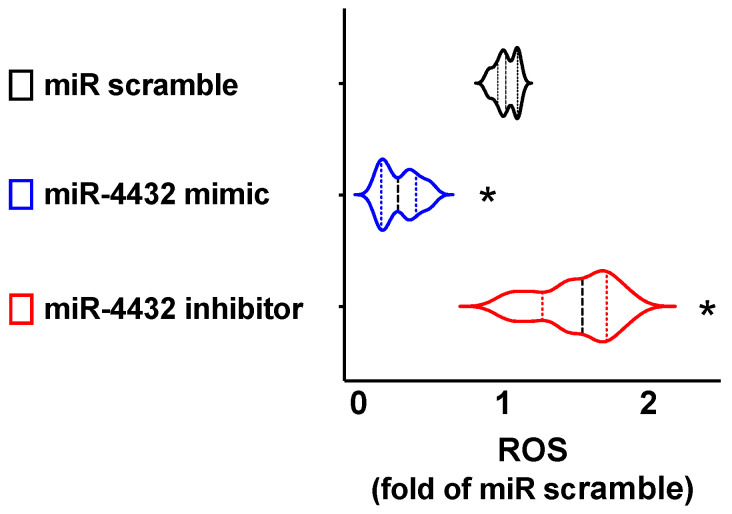
In hBMECs, the production of mitochondrial ROS (reactive oxygen species) was significantly diminished by miR-4432 mimic and increased by miR-4432 inhibitor. Mitochondrial ROS generation induced by Ang II (200 nMol, 4 h) was quantified using MitoSOX Red in hBMECs that had been transfected for forty-eight hours with the miRs indicated in the figure. All the assays were carried out at least in triplicate; the violin plots show the median (dashed line) and the quartiles (dotted lines); *: *p* < 0.01 vs. miR-scramble.

**Figure 8 biology-12-00459-f008:**
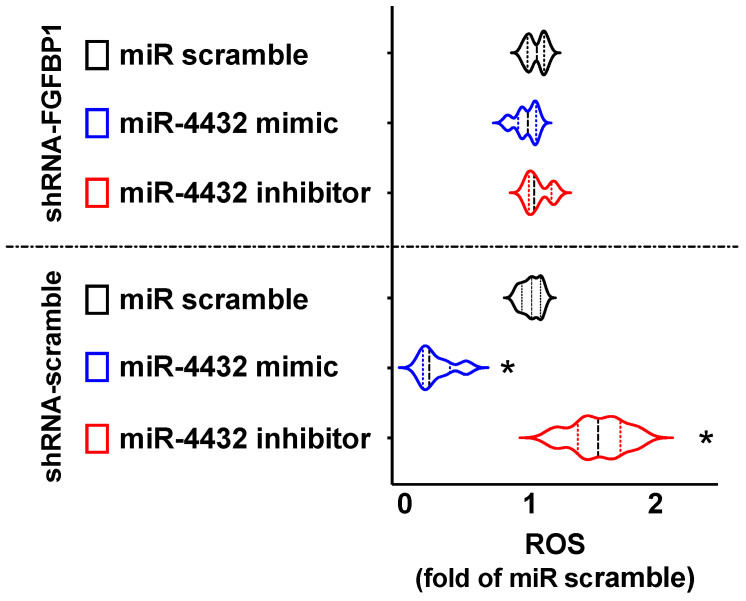
Mitochondrial ROS generation induced by Ang II in hBMECs was not affected when measured after having silenced FGFBP1 (top), whereas it was significantly blunted by miR-4432 mimic and increased by miR-4432 inhibitor when the cells had been treated with a shRNA scramble. All the assays were carried out at least in triplicate; the violin plots show the median (dashed line) and the quartiles (dotted lines); *: *p* < 0.01 vs. miR-scramble.

**Table 1 biology-12-00459-t001:** Primer sequences used for RT-qPCR assays.

Genes	F/R	Sequence (5′-to-3′)	bp
**FGFBP1**	** *Forward* **	GG AGG AGC TGT GAG TAA CGT	113
	** *Reverse* **	TG TCA GGT AGA GTG CAA GGG	
**GAPDH**	** *Forward* **	GG CTC CCT TGG GTA TAT GGT	94
	** *Reverse* **	TT GAT TTT GGA GGG ATC TCG	

FGFBP1 stands for fibroblast growth factor binding protein 1; GAPDH stands for glyceraldehyde-3-phosphate-dehydrogenase; bp indicates base pairs.

## Data Availability

All the data supporting the reported results are contained within this article and its Appendix A.

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
