# Peer review of "miR-4432 Targets FGFBP1 in Human Endothelial Cells"

_biology, 2023, doi:10.3390/biology12030459_

Round 1

Reviewer 1 Report

The study by Avvisato et al reports on the effects of miR-4332 on fibroblast growth factor binding protein 1 (FGFBP1) in endothelial cells in vitro. The authors provide data on the mRNA and protein level and show effects of the FGFBP1 level manipulation to mitochondrial reactive oxygen species production.

The manuscript is well-written and concise and it provides sufficient data. However, there are still some minor points

1.      Indicate more clearly what the authors call ‘miR-4332 mimic’ and what ‘miR-4332 inhibitor’. The most appropriate places seem to be either paragraph 4.2 or Fig1 or both.

2.      Whether wild-type miR-4332 is expressed in endothelial cells? Is there any available data in this regard? Please, reflect that in the text.

3.      Can the authors make any estimate regarding what transfection % was attained with the method used by the authors?

4.      Self-citations are 17 out of 92 references = 18.5%. Are all of the 92 cited references, including self-citations, fundamentally needed for a concise manuscript like this one?

5.      In Table1, there is no yellow highlight, which the legend refers to.

Author Response

R: Thank you for the word of appreciation toward our work.

1: Thank you for your comment: the definitions of miR-mimic and inhibitor have been added, as requested.

2: Thank you for your insightful comment: we are showing that miR-4432 is actually expressed in human endothelial cells (Figure 2).

3: We now specify in the revised version of the manuscript that we attained 66% transfection efficiency with Lipofectamine-RNAiMAX

4: We reduced both.

5: We can see the yellow highlighting; most likely there was an issue when the publisher transformed the Word file as .pdf

Author Response

R: We thank this Reviewer for her/his time spent in reviewing our paper.

1: We thank this Reviewer for the insightful remark: we are now showing that silencing FGFBP1 blunts the effects of miR-4432 (Figure 7).

2: We reduced both.

3: We re-organized the order of the figures.

4: The supplementary figure has been uploaded as .pdf using the link provided by the publisher.

5: We now clarify that the 2 lanes represent biological replicates; the entire assay shown in the representative blots (2 lanes per condition) was repeated 3 times and is quantified in panel B).

6: We respectfully believe that the fact that miR-4432 inhibitor is associated with an increased expression of FGFBP1 is actually corroborating our hypothesis: indeed, in absence of a functional miR-4432 that represses its transcription, the expression levels of FGFBP1 are supposed to increase because the inhibition has been relieved.

Round 2

Reviewer 2 Report

The autors addressed the concerns and revised the manuscript accordingly. The revised manuscript can be accepted for publcation.